# Structure and Hierarchy of Influenza Virus Models Revealed by Reaction Network Analysis

**DOI:** 10.3390/v11050449

**Published:** 2019-05-16

**Authors:** Stephan Peter, Martin Hölzer, Kevin Lamkiewicz, Pietro Speroni di Fenizio, Hassan Al Hwaeer, Manja Marz, Stefan Schuster, Peter Dittrich, Bashar Ibrahim

**Affiliations:** 1Ernst-Abbe University of Applied Sciences Jena, Carl-Zeiss-Promenade 2, 07745 Jena, Germany; stephan.peter@eah-jena.de; 2Bio Systems Analysis Group, Department of Mathematics and Computer Science, University of Jena, Ernst-Abbe-Platz 2, 07743 Jena, Germany; pietros@gmail.com; 3RNA Bioinformatics and High-Throughput Analysis, Friedrich Schiller University Jena, Leutragraben 1, 07743 Jena, Germany; martin.hoelzer@uni-jena.de (M.H.); Kevin.Lamkiewicz@uni-jena.de (K.L.); manja@uni-jena.de (M.M.); 4European Virus Bioinformatics Center, Leutragraben 1, 07743 Jena, Germany; 5Mathematics and Computer Applications Department, Al-Nahrain University, Al-Jadriya, Baghdad 10072, Iraq; hassan1167@yahoo.com; 6Chair of Bioinformatics, Matthias-Schleiden-Institute, University of Jena, Ernst-Abbe-Platz 2, 07743 Jena, Germany; stefan.schu@uni-jena.de

**Keywords:** chemical organization theory, influenza A, virus dynamics modeling, complex networks analysis

## Abstract

Influenza A virus is recognized today as one of the most challenging viruses that threatens both human and animal health worldwide. Understanding the control mechanisms of influenza infection and dynamics is crucial and could result in effective future treatment strategies. Many kinetic models based on differential equations have been developed in recent decades to capture viral dynamics within a host. These models differ in their complexity in terms of number of species elements and number of reactions. Here, we present a new approach to understanding the overall structure of twelve influenza A virus infection models and their relationship to each other. To this end, we apply chemical organization theory to obtain a hierarchical decomposition of the models into chemical organizations. The decomposition is based on the model structure (reaction rules) but is independent of kinetic details such as rate constants. We found different types of model structures ranging from two to eight organizations. Furthermore, the model’s organizations imply a partial order among models entailing a hierarchy of model, revealing a high model diversity with respect to their long-term behavior. Our methods and results can be helpful in model development and model integration, also beyond the influenza area.

## 1. Introduction

Influenza is an infectious respiratory disease, annually infecting 5–15% of the human population and causing epidemics that result in 3–5 million severe cases with 300,000–500,000 deaths each year [1]. The annual recurrence of epidemics is caused by the continuous alteration of seasonal influenza viruses, which enables them to efficiently escape the immune system even due to previous infections or vaccinations [2]. The major burden of disease in humans is caused by seasonal influenza A (IAV) and influenza B viruses, causing symptoms varying from mild respiratory disease characterized by fever, sore throat, headache and muscle pain to severe and in some cases lethal pneumonia and secondary bacterial infections [3].

The long-term spread of influenza viruses in the human population and the acute nature of influenza virus epidemics is driven by the global movement of these viruses. Differences in seasonal epidemics caused by influenza viruses are mainly driven by differences in the rates of virus evolution. The single-stranded RNA segments of influenza viruses, which are located inside the virus particle (or virion), evolve rapidly and thus can escape the host’s immune response very efficiently.

Several ordinary differential equations (ODEs) models have been developed to provide insight into within-host dynamics of influenza A virus infections (for reviews, see [4,5,6,7,8]). These models work in a time scale of days and describe the concentration dynamics of target cells, immune system components, viral load, and sometimes co-infecting pathogens. The models differ in terms of complexity and state space dimensions, which are between 3 and 15 for the models examined here. While the low-dimensional models can be analyzed completely and in a straightforward way (e.g., by calculating their fixed points and stability analysis), the characterization of the entire behavioral spectrum of complex models is more difficult (see for example [9]).

Here, we present an approach to understand the overall structure of these models that allows them to be related to each other in a simple way. To this end, we apply chemical organization theory [10,11] to obtain a hierarchical decomposition of each model into chemical organizations. A chemical organization is a sub-set of species (i.e., dimensions or model components, like, uninfected cells or viruses) that cannot generate any other species (property of closure) and that can self-maintain its own species, i.e., any species consumed by a process within the organization can be regenerated by a process within the organization. The organizations of an ODE model are rigorously related to its long-term dynamics in the following way: Given a stationary state of the ODE model, the set of species with strictly positive concentrations must be an organization [10]. The same is true for all practically relevant periodic and chaotic attractors [12]. Note that the advantage of this approach is that decomposition into organizations is based solely on the model structure (i.e., reaction rules) and thus is independent of kinetic details, like rate constants. The relation between measured data, ODE model, and organizations is depicted in Figure 1.

By applying the method to twelve models of influenza A virus infection, we found different types of model structures ranging from two to eight organizations. Furthermore, the models’ organizations imply a partial order among models. The resulting hierarchy of models can help to select a suitable model for certain data or serve as a framework for further model development.

We provide reaction network files for all models and a software tool for computing their organizations (https://github.com/stephanpeter/orgsflu).

## 2. Materials and Methods: Procedure for the Organizational Analysis

To illustrate our method, we follow a basic ODE model of influenza dynamics, namely the target cell limited model by Baccam et al. [13], called Baccam Model in the following. Note that we refer throughout this paper to a model by the first author’s name of the respective publication.

The Baccam Model is based on in vivo experimental data and contains three variables: the number of susceptible and uninfected target (epithelial) cells *T*, the number of infected cells *I*, and the number of infectious-viral titer *V*. The dynamical behavior of the variables is given by the ODEs shown in Figure 2a. That is, target cells become infected and thus converted to infected cells at a rate βTV, infected cells die spontaneously at rate δI, virus proliferates at a rate pI and dies at a rate cV. The parameters β, δ, *p* and *c* are, as usual, positive real numbers (cf. [13] for actual values).

Throughout this paper, the following coloring scheme for particular classes of species is used to improve readability:Uninfected (target) cells or those resistant/refractory to infection are marked in blue, e.g., *T*.Infected cells, partially or latently infected cells, and viruses are marked in magenta, e.g., *I* and *V*.Species belonging to the active immune response are marked in green. It is worth noting that the first two models analyzed in this paper [13,14] do not explicitly have immune system components.Bacterial co-infection species are marked in orange. These species are only occurring in Smith’s model [15].Text referring to any other species is marked in black, e.g., transient target cell states, passive immune system, or dead cells.

For simplicity, the models’ variable names also denote the related *species*. For example, *V* denominates not only the number of viruses in the ODE model (Figure 2a), but also the virus itself (e.g., Figure 2b).

### 2.1. Deriving the Reaction Network from the ODE System

In a first step, we need to obtain the reaction network underlying the ODE model. A reaction represents, for example, a cell infection by a virus, the generation of new viruses from an infected cell or the spontaneous death of a cell. The reaction rules can be derived from the ODEs in a straightforward way [16]. This step can also be performed by an online tool presented by Soliman and colleagues [16]. Note that in modeling one first creates the network and then derives the ODEs. For our analysis, we have to take the other direction.

For this purpose, we have to investigate the kinetic terms (kinetic laws) of the ODE (Figure 2a):The term βTV represents the reaction $R_{1}:\;T+V\to I+V$, which in turn denotes the transformation of an uninfected target cell *T* to an infected cell *I* catalysed by the virus *V*.The terms −δI and −cV represent reactions $R_{2}:\;I\to \emptyset$ and $R_{4}:\;V\to \emptyset$ which are the outflow of infected cells *I* resp. virus *V*.The term pI represents the reaction $R_{3}:\;I\to I+V$ which is the production of viruses *V* catalysed by infected cells *I*.
The set of species S={T,I,V} together with the set of reactions $R=\{R_{1},R_{2},R_{3},R_{4}\}$ constitute the so-called *reaction network*
(S,R) associated with the model. The set of reactions together with their kinetic parameters are depicted in Figure 2c. Note that for clarity we use different types of underlining to highlight certain recurring kinetic terms in the ODE:Single underline for the transformation of uninfected cells into infected ones by the action of viruses.$\underset{=}{\text{Double underline}}$ for kinetic terms involving interferon.

We call the species on the left-hand side (LHS) of a reaction *R support* of *R* and write supp(R), e.g., $supp\left(R_{1}\right)=\left\{T,V\right\}$ (see Figure 2d). Analogously, we call the set of species occurring on the right-hand side (RHS) of a reaction *R products* of *R* and denominate this set by prod(R), e.g., $prod\left(R_{1}\right)=\left\{I,V\right\}$.

Furthermore, let us recall the *stoichiometric matrix*
$N=(n_{ij})$ of a reaction network [17]. The element $n_{ij}$ in the *i*-th line and *j*-th column of N denotes the net-production of the *i*-th species in reaction $R_{j}$. The net-production $n_{ij}$ is the difference between the number of occurrences (stoichiometric coefficient) of species *i* in the RHS of reaction $R_{j}$ minus the number of occurrences of species *i* in the LHS of reaction $R_{j}$. For example, $n_{21}=1-0=1$ because the second species (*I*) appears in the first $R_{1}$ once as a reactant in the support of $R_{1}$ (LHS) but does not appear in $R_{1}$ as a product (RHS). For our example in Figure 2, the stoichiometric matrix becomes:(1)$$\begin{array}{c} N=\left(\begin{array}{cccc} -1 & 0 & 0 & 0\\ 1 & -1 & 0 & 0\\ 0 & 0 & 1 & -1 \end{array}\right). \end{array}$$

### 2.2. Computing the Organizations from the Reaction Network

From the reaction network, we can compute the (chemical) *organizations* of the model. Each organization is a subset of species that is *closed* and *self-maintaining* [10,18]. In the following, let S⊆S be a subset of species and *n* be the total number of reactions of the reaction network (n=4 in our example).

We call *S **closed*** if and only if all reactions R∈R with supp(R)⊆S fulfill prod(R)⊆S too [10,18]. This means that the products of a reaction *R* with support in *S* are also in *S*. In other words, no species outside of *S* can be produced by the reactions “running on” *S*. As an example, we assume S={T,I}. The reactions with support in *S* are $R_{2}$ and $R_{3}$. However, $R_{3}$ produces species *V*, which is not in *S*. Thus, *S* is not closed.

We call a vector $v\in R^{n}$
*flux vector for S* if and only if it fulfills
(2)$$\begin{array}{c} v_{j}\left\{\begin{array}{cc} >0, & \mathrm{if}\;\;\;supp(R_{j})\subseteq S,\\ =0, & \mathrm{if}\;\;\;supp(R_{j})⊈S. \end{array}\right. \end{array}$$
Thus, all flux vectors for *S* have in common that those components are strictly positive which correspond to reactions that can run on *S*, while all other entries are zero. As an example, consider $S_{Example}=\left\{T,I\right\}$ again. We know that the reactions $R_{2}$ and $R_{3}$ can "run on" it, i.e., they have support in $S_{Example}$. Thus, $v_{1}={\left(0,1,3,0\right)}^{T}$ or $v_{2}={\left(0,5,2,0\right)}^{T}$ are example flux vectors for $S_{Example}$.

We call *S **self-maintaining*** if and only if there exists (at least one) flux vector $v\in R^{n}$ for *S* that fulfills
(3)$$\begin{array}{c} N\cdot v\geq 0, \end{array}$$
i.e., ${\left(N\cdot v\right)}_{i}\geq 0$ for all i=1,…,n, where *n* is again the total number of reactions [10,19,20,21]. Roughly speaking, if *S* is self-maintaining, it has the potential to sustain the amount of its species above a certain level. Our example set $S_{Example}$ is not self-maintaining because ${\left(N\cdot v\right)}_{2}<0$ for all flux vectors for $S_{Example}$. That is, species number 2 (the infected cells *I*) can not be maintained by this set.

As mentioned in the beginning of this section, we call *S* an *organization* if and only if it is both closed and self-maintaining. Clearly $S_{Example}$ is not an organization, as it has neither of these properties. In Figure 2d, the so-called *Hasse diagram* of organizations of this model can be seen. In it, two organizations are linked by a line if the lower one is a subset of the upper one and there is no organization in between. The Hasse diagram for the Baccam Model contains only the two organizations $O_{1}^{B}=\emptyset$ and $O_{2}^{B}=\left\{T\right\}$ (see Figure 2d). The superscript “B” stands for Baccam Model and with the subscripts we refer to the organizations within a model. The organization $O_{2}^{B}$ represents an organism without any influenza A virus infection. Note that there is no organization with all species, i.e., representing the infected body.

### 2.3. The Role Organizations Play in the Dynamics

Given a fixed point *x* of an ODE describing a reaction system, the set of species with strictly positive concentrations in *x* is an organization. This is shown in [10]. This in turn means that, if a subset of species is not an organization, then the system does not have a fixed point with exactly the chosen subset of species. This is not only true for fixed points but also for other attractors [12]. Attractors are those states that a system approaches in the long-run and once reached never leaves anymore. Besides converging towards a fixed point, the long-term behavior can be also periodic oscillations or chaotic trajectories. In particular, it was shown that the long-term behavior of the system always tends at least to one organization [12]. Thus, organizations rule the long-term behavior of such dynamical reaction system. Note that the case of a system tending towards a fixed point is included in this statement as a special case. For example, the *simulation* results of the Baccam model (Figure 2a and Ref. [13]) suggest that after about two to three days the species begin to decay to finally arrive in an organization namely the empty set.

## 3. Results and Discussion

In the literature, there exist several mathematical models of IAV dynamics that are derived from experimental data, reviewed in Refs. [4,5,6,7]). These models differ in their complexity, e.g., the number of reactions and the number of species, depending on the available experimental data used for parameter fitting and questions to be answered. For example, models can include eclipse phases, an innate immune response, or an adaptive immune response.

After having exemplified our method in Section 2 by an analysis of the basic target cell limited model by Baccam et al. [13], we present now the full analysis for eleven additional more detailed influenza models of IAV dynamics, with up to 15 variables (species) and 45 reactions (cf. Table 1 for an overview at the end). Note that for our analysis we abstract from kinetic details, that is, the organizations are independent of particular settings of parameter values.

### 3.1. Target Cell Limited Model by Miao et al. (Miao Model, M, 2010)

The models by Miao et al. [14] are designed to fit experimental in vivo data from mice [6,7]. The first one ([14], Equation (Equation 1)) depends on measured time-series. The second one ([14], Equation (Equation 2)) is a simplified version of the first one, neglecting the terms depending on those time-series and still leading to a good fit within the first five days after infection [14]. Thus, we analyze this second model (Miao Model).

Compared to the basic Baccam Model, the Miao Model (Figure 3a) has the same three variables (named differently) and one additional reaction, $E_{P}\to 2E_{P}$. This reaction represents the self-replication of target cells $E_{P}$ taking place at a rate $\rho_{E}E_{P}$. The full set of reactions can be found in the Appendix A (Figure A2).

In the Hasse diagram of organizations (Figure 3b), a new “full” organization $O_{3}^{M}$ appears, which contains all three species. Thus, organization $O_{3}^{M}$ reflects the slight difference between the two models: in the Baccam Model, uninfected target cells *T* are only the susceptible ones and can not increase in number, but in the Miao Model uninfected cells $E_{P}$ are reproduced repeatedly by the organism. Thus, in the Baccam Model, infection is limited inherently by the limited number of uninfected target cells, while in the Miao Model the limitation of an infection in time and number of infected cells and viruses depends on other mechanisms:

### 3.2. Target Cell Limited Model with Delayed Virus Production (Baccam II Model, Ba2, 2006)

The Baccam II Model [13,22] contains one more species than the Baccam Model presented in the methods section above. That is, there are now *two* types of infected cells: those which do not yet produce viruses $I_{1}$ and those which actively produce viruses $I_{2}$. In addition, there is only one new reaction, which transforms infected cells of type $I_{1}$ into type $I_{2}$ at rate $kI_{1}$ (Figure 4a). However, the Hasse diagram of organizations remains the same when compared with the basic Baccam Model [13].

### 3.3. Innate and (Simple) Adaptive Immune Response (Pawelek Model, P, 2012)

The Pawelek Model [23] contains 11 parameters and was designed to fit in vivo experimental data of horses [6,7]. The model has five variables and nine reactions. Like the basic Baccam Model, it contains uninfected target cells (*T*), infected cells (*I*), and viruses (*V*). Furthermore, there are two new species: interferon (*F*) and uninfected cells that are refractory to infections (*R*) because of the antiviral effect induced by interferons.

Investigating the reaction network (Figure A4 in the Appendix A) derived from the differential equations (Figure 5a), we can see that, like in the basic Baccam Model, self-replication of uninfected cells *T* is missing. However, due to the two new species *R* and *F*, we have five new reactions, which are neither included in the Baccam Model nor in the Miao Model. One of these five reactions is the spontaneous decay of interferon *F* at a rate dF. The other four new reactions describe interactions between different species:The rate term ϕFT represents the transformation of uninfected target cells to refractory cells catalysed by interferon.The reverse shift back from refractory to simple uninfected cells is represented by the term ρR.Furthermore, infected cells are deleted by the action of interferon at a rate κIF.Interferon is produced in the presence of infected cells at a rate qI.

Even though we have more species and more reactions, we get the same small pattern of organizations as in the basic Baccam Model (Figure 5b). Both models have in common that there is no self-replication of target cells. This might be one reason for the missing of other and bigger organizations which could contain species related to infection and/or immune response. This in turn means that, like the Baccam Model, this model implicitly treats infections and immune responses as phenomena that can only appear in a limited (transient) time span. The Hasse diagram of organizations (Figure 5b) tells us that the system necessarily tends towards a state of healthiness, which is represented by the organizations $O_{1}^{P}=\left\{\right\}$ and $O_{2}^{P}=\left\{T\right\}$, showing absence of infection and immune response.

### 3.4. A Model Including Bacterial Co-Infection (Smith Model, Sm, 2016)

The Smith Model [15] contains 15 parameters and compared to experimental in vivo data from mice. It has five variables and 12 reactions. Like in the previous models, we have susceptible target cells (*T*) and viruses (*V*). Note that *T* is only consumed in this model but not produced. Contrarily to previous models, we have two kinds of infected cells ($I_{1}$ and $I_{2}$) here as well as bacteria (*P*). Bacteria *P* represent *bacterial co-infection* during or after virus infection. Infection is modeled as a transformation of susceptible target cells *T* into infected cells $I_{1}$ catalyzed by viruses *V* (see underlined terms in Figure 6a). Infected cells $I_{1}$ in turn spontaneously transform into $I_{2}$ at rate $kI_{1}$. Only infected cells $I_{2}$ produce viruses *V* at a rate $pI_{2}$. Furthermore, infected cells $I_{2}$ produce viruses *V* together with bacteria *P*. Bacteria *P* are self-replicating (rate term rP). Viruses *V* are the only species influencing bacteria *P*.

Figure 6b shows the Hasse diagram of organizations. The smallest one is the empty set. The biggest one is $O_{4}^{Sm}$, which contains susceptible target cells *T* and bacteria *P*. It represents an organism without viral but with bacterial infection. Between those two extreme organizations, we find $O_{2}^{Sm}=\left\{\begin{array}{c} T \end{array}\right\}$ and $O_{3}^{Sm}=\left\{\begin{array}{c} P \end{array}\right\}$. Thus, $O_{2}^{Sm}$ represents the healthy organism without any infection. $O_{3}^{Sm}=\left\{\begin{array}{c} P \end{array}\right\}$ could mark the state after a viral-bacterial co-infection: after viral infection and because of the death of all target cells as well as all viruses only bacteria remain.

### 3.5. Innate and Adaptive Immune Response (Handel Model, Ha, 2009)

The Handel Model [24] contains eight parameters and was designed to fit experimental in vivo data from mice [6,7,25]. It has seven variables (see Figure 7) and only 12 reactions (see Figure 7a):*Infection* is catalyzed by viruses *V* and transforms uninfected cells *U* to latently infected cells *E* and viruses *V* are consumed thereby. Latently infected cells *E* transform into infected cells *I* autonomously, which in turn transform into dead cells *D* autonomously too. Finally, the transformation of dead cells *D* into non-infected cells *U* closes the *circle*.The *remaining three species V*, *F* and *X* form an almost totally separate *subsystem* since the only interaction with the four species from the "circle" mentioned above is the catalysis of the infection by viruses *V*.The *interactions within the subsystem*
{V,F,X} consisting of viruses *V* and immune responses *F* and *X* are as follows:
–Viruses *V* catalyze the proliferation of *F* and *X*. In the Hernandez model, proliferation of interferon *F* is catalyzed by infected cells instead of viruses.–There is no direct interaction between innate immune response *F* and adaptive immune response *X*.–The adaptive immune response *X* deletes viruses directly. Innate immune response *F* inhibits the self-replication of the viruses which is represented by the denominator of the fraction $\frac{pI}{1+\kappa F}$. We ignore the inhibition because whether the rate is zero or not is independent of *F*.
The Hasse diagram Figure 7b shows five organizations. For the first time, it contains the empty set as well as the set of all species at the same time. Between these extremes, we find $O_{2}^{Ha}=\left\{U\right\}$ representing the healthy organism. The Baccam, Miao, and Pawelek models exhibit the same organization. Structurally, the Hasse diagram of the Handel Model is very similar to that of the Smith Model (Figure 6b). The first reveals an autonomy of the adaptive immune response *X*, whereas the latter does this same for bacteria *P*.

#### Temporal Dynamics

For the Handel Model, we perform dynamical simulation in order to show how the organizational hierarchy helps also to understand transient short-term behavior. We start at t=−20d with an uninfected state, i.e., $7\times 10^{9}$ uninfected cells. After 20 days, at t=0, we add $10^{4}$ virus particles. The resulting seven-dimensional trajectory in state space is shown in Figure 8. Projecting this trajectory to organizations results in a more abstract view of the dynamics, shown as a dashed curved arrow in Figure 7b. The system starts in organization $O_{2}^{Ha}$ (uninfected organization), moves after adding virus particles at t=0 into organization $O_{5}^{Ha}$ (infected organization with immune response), and drops after 37 days into organization $O_{4}^{Ha}$ (immune response active, virus absent).

The projection of a state *x* to an organization *O* follows the procedure suggested by Dittrich and Speroni d.F. [10]: First, we generate a set *S* of those species whose concentration is greater than a particular threshold (here: $10^{0}=1$). Then, we generate the closure of this set by adding all species that can be produced from the set. Finally, we take the largest organization *O* that is a subset of that closure. For example: At t=0 by adding viruses to the system, we have S={U,V}, whose closure is the set of all species, which is also an organization, here; thus, the state at t=0 is projected to organization $O_{5}^{Ha}$. At t=60d, we have S={U,X,D}, whose closure is again {U,X,D} and the largest organization contained is $O_{4}^{Ha}=\left\{U,X\right\}$. Thus, as can be seen in Figure 8, the system state is projected to organization $O_{4}^{Ha}$, in which it stays for t>37d.

### 3.6. Innate Immune Response and Resistance to Infection (Hernandez Model, He, 2012)

The 13 parameters of the Hernandez Model [26] were fitted to data from many different sources. The model contains seven variables and 16 reactions (see Figure 9). The species refer to viruses (*V*), interferon (*F*) and natural killer cells (*K*) as well as four types of respiratory tract epithelial cells: healthy/uninfected ($U_{H}$), partially infected ($U_{E}$), infected ($U_{I}$) and resistant to infection ($U_{R}$). Compared to the Pawelek Model, there are two qualitatively new species: partially infected cells $U_{E}$ and killers *K*.

Next, we state some remarks about the reactions:There is an *infection* reaction catalyzed by viruses like in all previous models but with one difference: during infection, healthy cells $U_{H}$ first transform to *partially infected* cells $U_{E}$ and only after that they transform spontaneously to infected cells $U_{I}$ at a rate $k_{e}U_{E}$.*Interferon* catalyzes the transformation of healthy cells to resistant cells $U_{R}$, like in the Pawelek Model. However, in the Pawelek Model, interferon removes infected cells. Here, interferon’s production is catalyzed by infected cells $U_{I}$ at a rate $a_{I}U_{I}$. There is no further influence of interferon on any other species.Infected cells are removed by natural *killers K*, which also delete partially infected cells in this model. The production of *killers K* is catalyzed by infected cells $U_{I}$ at a rate $\Phi_{K}U_{I}$.Note that here we have an constant *inflow of healthy cells*
$U_{H}$ at a rate $S_{H}$ (first differential equation). Thus, healthy cells cannot converge to zero.
The Hasse diagram of organizations consists only of two organizations (Figure 9b). For the first time, the empty set is not an organization because the empty set is not closed due to a constant inflow of healthy cells $U_{H}$ and killers *K*, represented by the reaction $\emptyset \to U_{H}$ and ∅→K, respectively. The smallest organization $O_{1}^{He}=\left\{U_{H},K\right\}$ can be regarded as a state of healthiness. Contrarily, the second organization $O_{2}^{He}$ contains all species (as in the Miao Model) and therefore can be interpreted as the infected organism exhibiting immune response to infection.

### 3.7. A Model with More Detailed Immune Response (Cao Model, C, 2015)

The Cao Model [27] consists of 20 parameters and has been derived by referring to experimental data from ferrets. The model has seven variables and 26 reactions. Like in most of the previous models, we have (uninfected) target cells (*T*), infected cells (*I*), viruses (*V*), resistant cells (*R*), and interferon (*F*). Furthermore, there are B cells *B* and antibodies *A*.

According to the ODE shown in Figure 10a, *B cells* are only influenced by viruses: viruses support the production of B cells (rate term $m_{1}V$), but the more B cells that are present, the more of them are destroyed by viruses again (term: $m_{1}VB$). B cells influence only one other species, namely antibodies *B*, which they produce. *Antibodies A* influence only one other species, namely viruses, which are destroyed by this reaction at rate μVA. Antibodies in turn are influenced by B cells positively and by viruses negatively.

There are three organizations in this model (Figure 10b): the empty set, the healthy organism without any infection and without any immune response ($O_{2}^{C}=\left\{T\right\}$) and the organization $O_{3}^{C}$ containing all species.

### 3.8. Innate Immune Response and Eclipse Phase (Saenz Model, Sa, 2010)

The Saenz Model [28] requires 12 parameters and was designed to fit experimental in vivo data from horses [6,7]. The model contains eight variables and 12 reactions (Figure 11a). There are no adaptive immune response, no dead cells, and no natural killer cells. However, the model contains viruses *V* and interferon *F*. There is an eclipse phase ($E_{1}$ and $E_{2}$) here as well as prerefractory and refractory cells. In particular, epithelial cells are represented by six species: susceptible (*T*), eclipse phases ($E_{1}$
*and*
$E_{2}$), infectious (*I*), prerefractory (*W*), and refractory (*R*). Thus, the new features are the inclusion of two eclipse phases and three steps for the transformation of uninfected cells to refractory cells.

The Hasse diagram of organizations is composed by four organizations: $O_{1}^{Sa}=\left\{\right\}$: the empty set; $O_{2}^{Sa}=\left\{T\right\}$: representing a healthy organism; $O_{3}^{Sa}=\left\{R\right\}$: there is no consuming reaction for refractory cells *R*; $O_{4}^{Sa}=\left\{T,R\right\}$: also representing a healthy organism that contains refractory cells maybe as the remains of a previous infection.

The Hasse diagram is very similar to that from of the Handel Model. There are only two differences:The “full” organization is missing here. For sure, one of the reasons is that there is no reaction producing susceptible cells *T*. Thus, when viruses *V* or interferon *F* are present susceptible cells *T* can not survive and the “full” organization neither.Adaptive immune response is replaced by refractory cells in the organizations here.

### 3.9. Focusing on Innate and Adaptive Immunity (Hancioglu Model, Hcg, 2007)

The Hancioglu Model [29] contains 28 parameters, 10 variables, and 44 reactions. It has not been mathematically fitted to data but has been designed to meet specific general criteria [6,7]. The ODEs (Figure 12a) describe the dynamics of the following 10 species: viruses (*V*), healthy cells (*H*), infected cells (*I*), interferon (*F*) and resistant cells (*R*). The remaining species are new: antigen presenting cells (*M*), effector cells (*E*), plasma cells (*P*) antibodies (*A*) and antigenic distance (*S*). There are no species for an eclipse phase in this model.

Looking at the reaction network (Figure A8), we can see again a reaction for *infection*, i.e., the transformation of healthy cells *H* into infected cells *I* catalyzed by viruses *V* at a rate $\gamma_{HV}VH$ (single underlined in Figure 12a). Furthermore, *interferon F* is produced catalytically by antigen presenting *M* and infected cells *I*, decays spontaneously at a rate $a_{F}F$, and is additionally removed when converting healthy cells *H* into resistant cells *R* by the reaction H+F→R at rate $b_{HF}FH$ (double underlined in Figure 12a).

The Hancioglu Model has three organizations (Figure 12b):The smallest one is $O_{1}^{Hcg}$, which contains all the species responsible for the *immune response*.$O_{1}^{Hcg}$ is a subset of $O_{2}^{Hcg}$, which additionally contains species *H* and *R*, representing the healthy organism without infection but with the immune response turned on.$O_{3}^{Hcg}$ is the “full” organization containing all the species of the models and thus representing the organism with infection and immune response.
Thus, all the organizations represent meaningful states of the organism. However, there is no organization that only consists of healthy cells without any infection and immune response. Note that almost all the previous models except for Hernandez have such an organization.

### 3.10. Model with Delay Differential Equations (Bocharov Model, Bo, 1994)

The Bocharov Model [30] contains 49 parameters and was designed to fit experimental in vivo data from humans [6,7]. It includes 10 variables and 51 reactions (Figure A9). Only here and in the Lee Model (below) we have differential equations with *delay*, i.e., some rates depend on variable values from the past (Figure 13a). Because the delay does not matter in a steady-state, we can also neglect the delay when analyzing the chemical organizations of a delay differential equation model.

Note that this is by far the oldest model analyzed here. The names of the variables are a bit particular when compared to those of the previously analyzed models. As in all the other models, we have viruses $V_{f}$ and infected cells *C*. Furthermore, there are destroyed epithelial cells *m* as in the Handel Model. All other species belong to the immune response. Note that only in this model there is no state variable for uninfected, healthy cells. Bocharov et al. represent these healthy cells implicitly by subtracting the amounts of infected-cells *C* and destroyed epithelial cells *m* from the initial total amount of target epithelial cells $C^{*}$. Since all the other models analyzed here have a related variable, we inserted the variable $U=C^{*}-C-m$ for uninfected cells together with its ODE to make the model comparable to the others.

Due to the fact that the majority of the species belongs to the immune response, this is the case for most of the reactions too. These species of the immune response form exactly the organization $O_{1}^{Bo}$, only macrophages $M_{V}$ are missing.

Similarly to the Hancioglus model, the smallest organization $O_{1}^{Bo}$ is an organization with immune response but without infection (*C*, $V_{f}$). There is only one further organization that contains only one more species than $O_{1}^{Bo}$, namely uninfected cells *U*. This organization we already found in three of the previous models. However, for the first time, there is no bigger organization in this model. Thus, virus infection is necessarily transient.

### 3.11. Complex Dual-Compartment Model (Lee Model, L, 2009)

The Lee Model [9] is the most complex model considered here. It contains 48 parameters and was designed with respect to experimental in vivo data from mice [6,7]. It has 15 variables and 37 reactions (Figure A10). Like in the Bocharov Model, Lee et al. apply delay differential equations.

Note that this model is the only one analyzed here that distinguishes between *lung compartment* and *lymphatic compartment*. There is one species representing uninfected (healthy) cells $E_{p}$ and three species for modelling *infection*: $E_{P}^{*}$, $D^{*}$ and viruses V. The remaining species belong to the *immune response*, colored black when naive to infection, while colored green when activated for infection.

Note that we write a species in the *organizations* in Figure 14b in bold text, if it is “new”, that is, not contained in neither of its subset organizations. The Hasse diagram contains eight organizations. The smallest one is $O_{1}^{L}=\left\{E_{P},D,H_{N},T_{N},B_{N}\right\}$ and contains exactly the uninfected cells as well as the naive part of the immune response. The biggest organization contains all species. Between these two “extreme” organizations are six further organizations containing different parts of the activated part of the immune response.

### 3.12. Hierarchy of Influenza A Virus Models

In order to construct a hierarchical map of all investigated models, we characterize a model by a signature of organizations, which is a set of organization types. For example, the signature of the Handel Model (Figure 7b) is the set {∅,X,X,XX,XXX}. An organization type like XX means that there is at least one organization that contains uninfected (target) cells (X) *and* species of the active immune response (X). The deviation of the signatures for all models is shown in Table 1. Note that we ignore species colored black. We include the empty set *∅* because this distinguishes models without any inflow from those that possess an inflow of some species.

Now, we can obtain a partial order among models by defining that a model A is smaller or equal to another model B (A ≤ B), if the signature of A is a subset of the signature of model B. For example, the Hernandez Model is smaller than the Lee Model because {XX,XXX}⊆{X,XX,XXX}. This partial order among models leads to a hierarchical map of models, which is visualized by a Hasse-diagram in Figure 15. Note that a model A that is smaller than a model B according to this partial order can possess more species and reactions than B.

In Figure 15, we can see that all models have organizations with uninfected, healthy cells (X). There are models that furthermore have infection (X) and/or immune response (X) in their organizations. There are exactly two models (Hancioglu and Hernandez Model) with immune response in all their organizations which means that these models implicitly assume immune response to be active all the time. Among the models neglecting immune response are those which have infection (Miao Model) or bacteria (X) (Smith Model) in their organizations and also those that do not (Baccam and Baccam II Model). For models involving immune response, the situation is more complex. There are those that only have healthy cells in their organizations (Pawelek and Saenz Models). This means that these models implicitly exclude infection and immune response from the long run and thus treat them as transient phenomena a priori. The Bocharov Model is the only one that exhibits only healthy cells and immune response in its organizations but no infection. The remaining five models include all kinds of species (except for bacteria of course) in their organizations.

By looking at the hierarchy of models, it becomes evident that there is space for more models. Above the Smith and Handel Model, there could be one in which virus infection as well as bacterial coinfection can be simultaneously persistent (“fully persistent models” denote such hypothetical models in Figure 15). Another extreme case would be a “fully-transient model” in which we have only transient dynamics and all species would finally tend to zero. Such a model would be the smallest one in our partial order of models (Figure 15).

The derived hierarchical map of models might be used to choose the most appropriate model for a particular domain and data set: The model should contain at least one organization for each set of species that were experimentally observed to survive in the long run. If there are several models with such organizations, the one with the smallest organizations might be chosen to provide maximum efficiency in modeling. Table 1 as well as Figure 15 might guide the selection process, complementing established quantitative selection methods, such as those using the area under the viral load curve [31].

## 4. Conclusions

By analyzing twelve published IAV immune system models, we have shown that we can compute, independently of quantitative kinetic data like rate constants or kinetic laws, all chemical organizations of a typical IAV model, which provides a hierarchical decomposition of the model and an overview of its potential long-term behavior.

It turned out that the derived organizations are meaningful with respect to the model’s domain. That is, the composition of species inside an organization can be related to a particular state of the organism, like “healthy”, “infected”, or “virus controlled by active immune response”, and it is possible to annotate organizations accordingly.

By deriving an organizational signature from a model’s organizations, we obtained a novel classification scheme and a hierarchy of models with respect to their qualitative long-term behavior. Although the classification via organizational signature is quite coarse-grained, the analysis revealed still a high diversity of models. That is, the models have different potentiality with respect to which variables persist in the long turn and which vanish. Furthermore, the hierarchy map of models contains various empty territories, suggesting space for potential future models.

We envision as a practical use that our method and results can help to select the right model for a particular situation, to relate other models to the present ones, to obtain an overview of the potential long-term dynamics of complex models, and to support model development, for example, by providing a rapid consistency check. Note that measured long-term as well as transient data can be explained with respect to organizations, by defining a projection from a system state to an organization, as demonstrated for the Handel model (Figure 7 and Figure 8).

In addition, our approach is not limited to IAV models, but can be directly applied to other viruses in the same way since their dynamics are similarly modeled by ODEs [32]. Furthermore, our approach is open to include other dynamically relevant components like treatment and vaccination strategies. Another task for future work would be to study the transient virus immune system in more detail, for example, by mapping the basin of attraction of each organization or by systematically analyzing the transition dynamics in-between organizations [33,34].

## Figures and Tables

**Figure 1 viruses-11-00449-f001:**
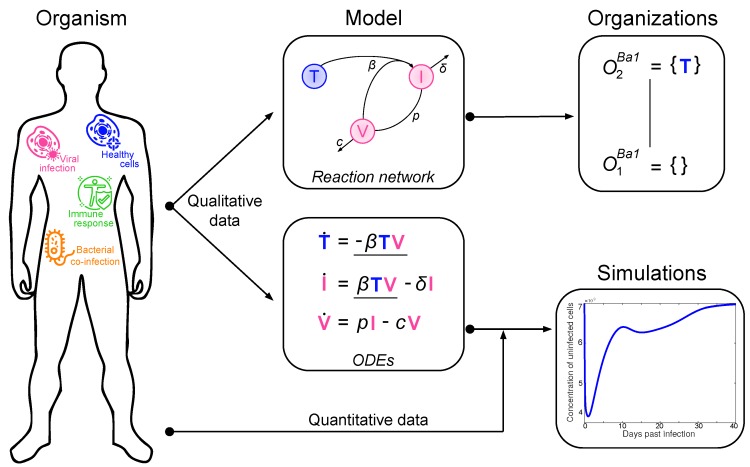
Relation between measured data, ordinary differential equations (ODE) model, and hierarchy of organizations.

**Figure 2 viruses-11-00449-f002:**
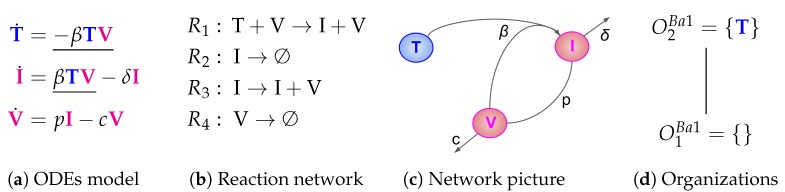
The **Baccam Model** [13] with three variables: uninfected (susceptible) target cells (T), infected cells (I) and infectious-viral titer (V).

**Figure 3 viruses-11-00449-f003:**
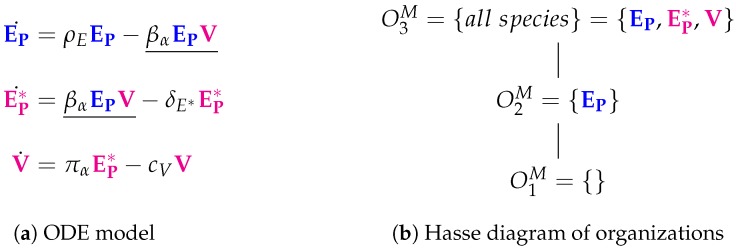
The **Miao Model** [14] with three variables: uninfected target cells ($E_{P}$), productively infected cells ($E_{P}^{*}$) and free infectious influenza viruses (V).

**Figure 4 viruses-11-00449-f004:**
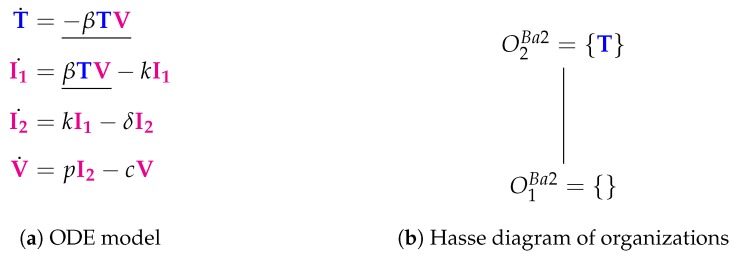
The **Baccam II Model** [13] with delayed virus production and four variables: uninfected (susceptible) target cells (T), infected cells not yet producing virus ($I_{1}$), infected cells actively producing virus ($I_{2}$) and infectious-viral titer (V).

**Figure 5 viruses-11-00449-f005:**
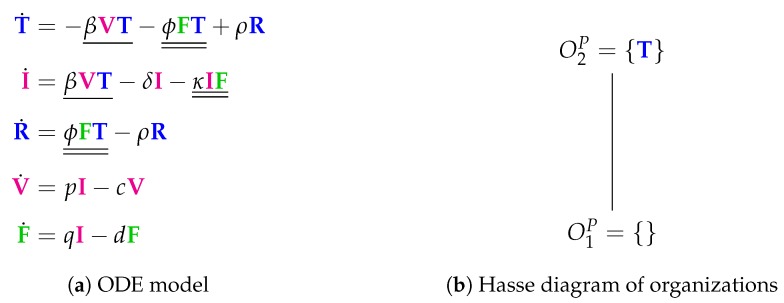
The **Pawelek Model** [23] with five variables: (uninfected) target cells (T), productively infected cells (I), uninfected cells refractory to infections (R), free viruses (V) and interferon (F).

**Figure 6 viruses-11-00449-f006:**
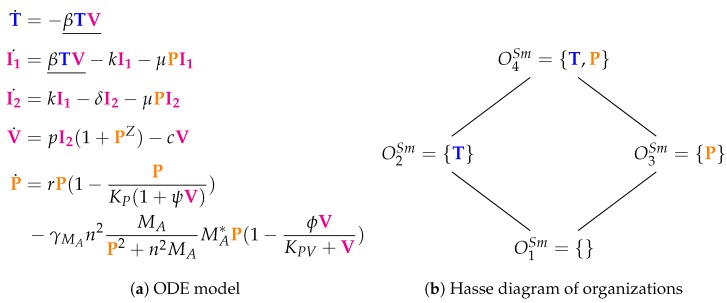
The **Smith Model** [15] with five variables: susceptible target cells (T), two classes of infected cells ($I_{1}$ and $I_{2}$), free viruses (V), and bacteria (P).

**Figure 7 viruses-11-00449-f007:**
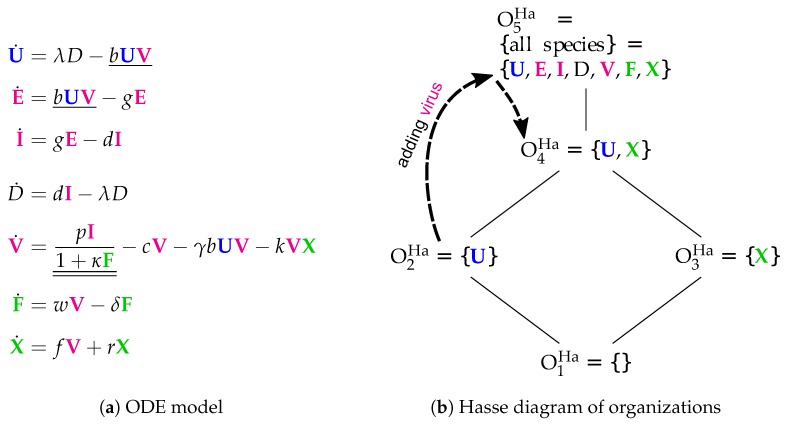
The **Handel model** [24] with seven variables: uninfected cells (U), latently infected cells (E), productively infected cells (I), dead cells (*D*), free viruses (V), innate immune response (F) and adaptive immune response (X). The dotted arrows denote the projection of the dynamics shown in Figure 8.

**Figure 8 viruses-11-00449-f008:**
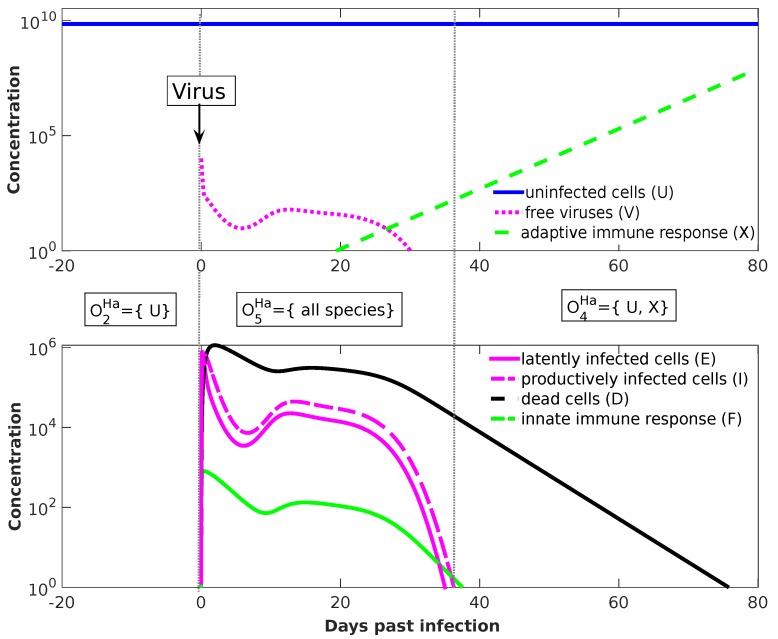
Temporal dynamics of the Handel model. By projecting the seven-dimensional trajectory to organizations (dotted arrows in Figure 7b) we find three phases: (Phase 1) Until day number 0, there are solely $7\times 10^{9}$ uninfected cells U in the system represented by the organization $O_{2}^{Ha}=\left\{U\right\}$. (Phase 2) At day 0, **infection** is simulated by adding $V\left(0\right)=10^{4}$ virus particles to the system. The resulting state {U,V} is projected to organization $O_{5}^{Ha}$ (all species). (Phase 3) Lastly, at day t = 37d past infection the system settles in the final organization, namely $O_{4}^{Ha}=\left\{U,X\right\}$, which is generated by the set {U,X,D} (see text). The values of the model parameters are (from [24]): λ=0.25, $b=2.1\times 10^{-7}$, g=4, d=2, $p=5\times 10^{-2}$, $\kappa =1.8\times 10^{-2}$, c=10, $\gamma =7.5\times 10^{-4}$, k=1.8, w=1, δ=0.4, $f=2.7\times 10^{-6}$, and r=0.3. Note that the number of uninfected cells U is not constant after infection as it may seem from the figure. In fact, after infection, the number of uninfected cells first decreases and than rises again [24].

**Figure 9 viruses-11-00449-f009:**
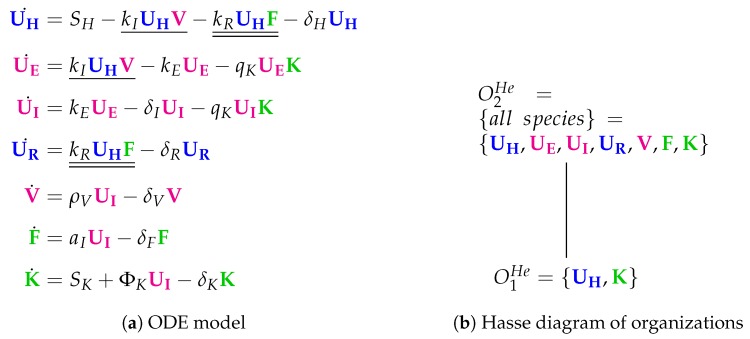
The **Hernandez Model** [26] with seven variables: healthy cells ($U_{H}$), partially infected cell ($U_{E}$), infected cells ($U_{I}$), cells resistant to infection ($U_{R}$), virus particles (V), interferon (F) and natural killers (K).

**Figure 10 viruses-11-00449-f010:**
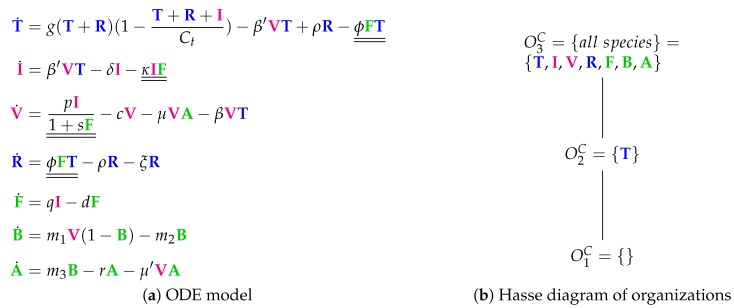
The **Cao Model** [27] with seven variables: target cells (T), infected cells (I), viruses (V), resistant cells (R), interferon (F), B cells (B), and antibodies (A).

**Figure 11 viruses-11-00449-f011:**
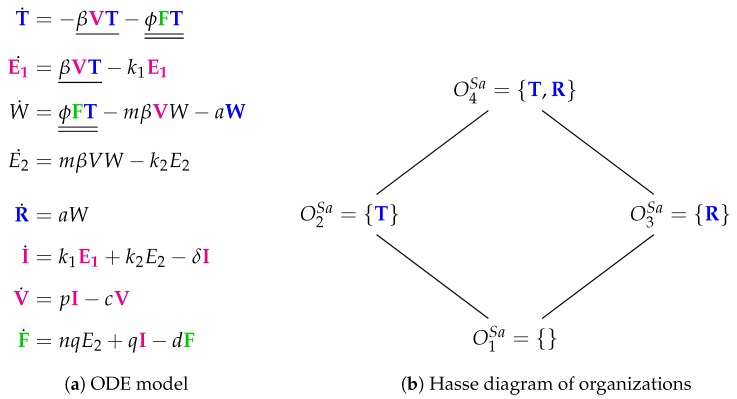
The **Saenz Model** [28] with eight variables: Epithelial cells in one of the states: susceptible (T), eclipse phase ($E_{1}$ and $E_{2}$), prerefractory (*W*), refractory (R) and infectious (I). The further variables are: virus cells (V) and interferon (F).

**Figure 12 viruses-11-00449-f012:**
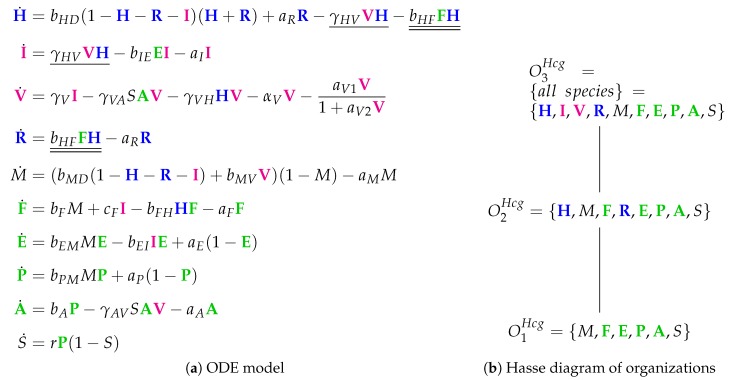
The **Hancioglu Model** [29] with 10 variables: viral load (V), healthy cells (H), infected cells (I), antigen presenting cells (*M*), interferon (F), resistant cells (R), effector cells (E), plasma cells (P), antibodies (A) and antigenic distance (*S*).

**Figure 13 viruses-11-00449-f013:**
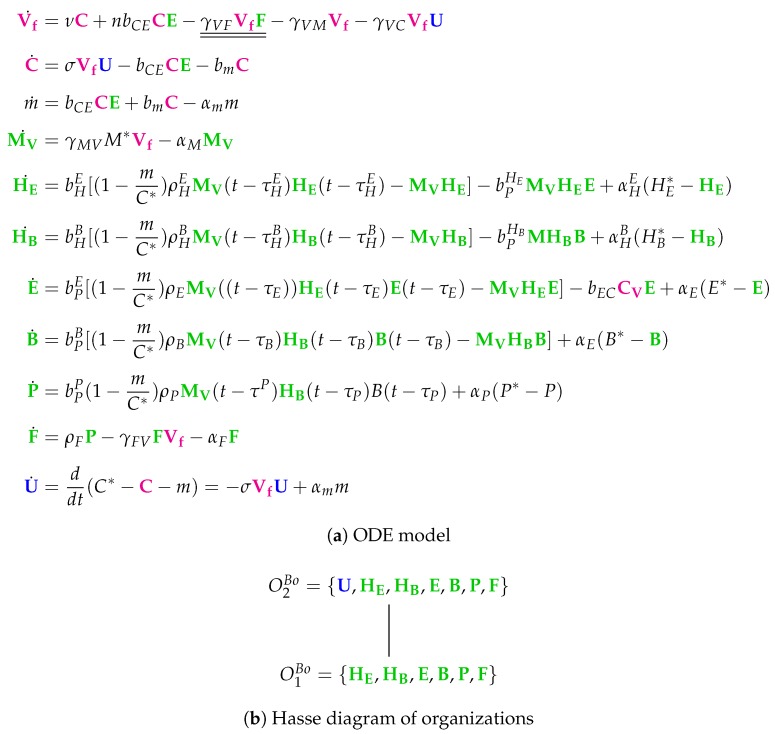
The **Bocharov Model** [30] with 10 variables: infective IAV particles ($V_{f}$), IAV-infected cells (C), destroyed epithelial cells (*m*), stimulated macrophages ($M_{V}$), activated helper T cells providing proliferation of cytotoxic T cells ($H_{E}$), activated helper T cells providing proliferation and differentiation of B cells B ($H_{B}$), activated CTL (E), B cells (B), plasma cells (P), antibodies to IAV (F), and uninfected epithelial cells (U). Note that, for clarity, we have added U as a state variable, which is only implicitly represented as $U=C^{*}-C-m$ in the original model by Bocharov et al.

**Figure 14 viruses-11-00449-f014:**
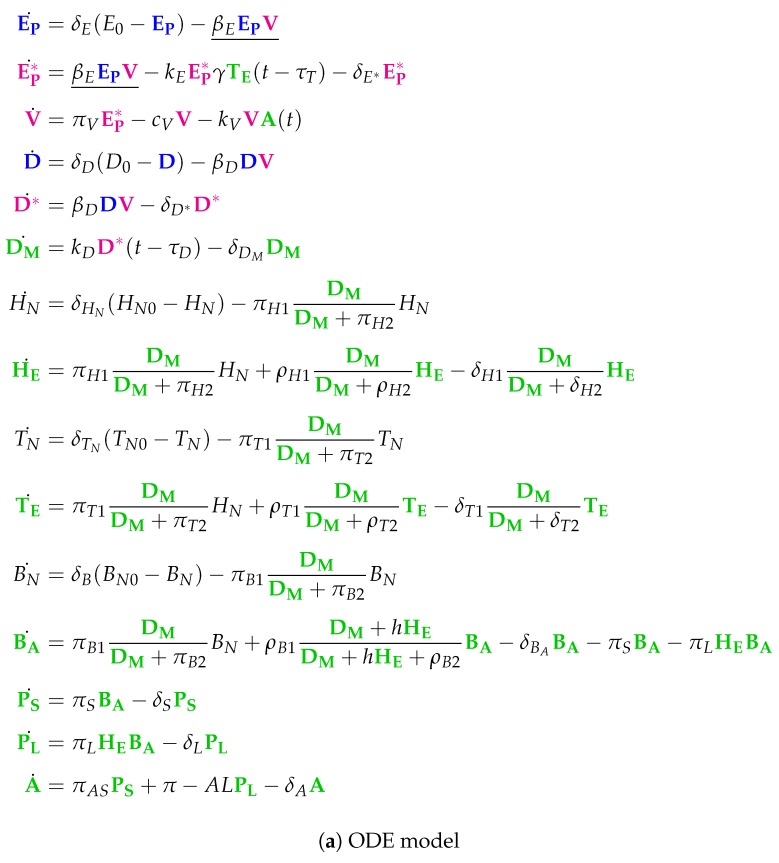
The **Lee model** [9] which contains 15 variables: uninfected epithelial cells ($E_{P}$), infected epithelial cells ($E_{P}^{*}$), virus titer ($EID_{50}/ml$) (V), immature dendritic cells (D), virus-loaded dendritic cells ($D^{*}$), mature dendritic cells ($D_{M}$), naive CD4+ T cells ($H_{N}$), effector CD4+ T cells ($H_{E}$), naive CD8+ T cells ($T_{N}$), effector CD8+ T cells ($T_{E}$), naive B cells ($B_{N}$), activated B cells ($B_{A}$), short-lived plasma (antibody-secreting) B cells ($P_{S}$), long-lived plasma (antibody-secreting) B cells ($P_{L}$) and antiviral antibody titer (A). Note that here we have colored green only those species representing the immune system when activated.

**Figure 15 viruses-11-00449-f015:**
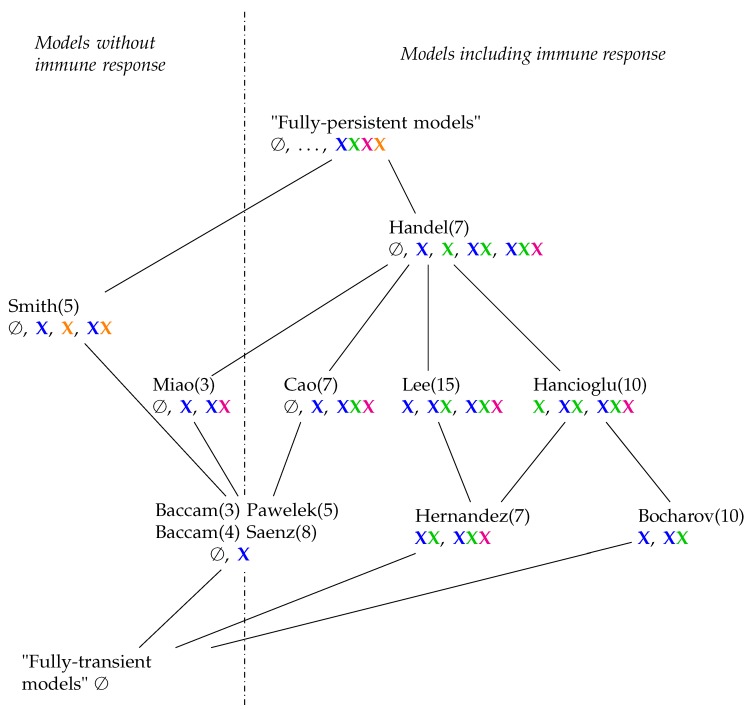
Hasse-diagram of the hierarchy of IAV models with respect to their long-term behaviour. In brackets (), we added the number of species of each model. Underneath (marked by colors) the kinds of species contained in the organizations belonging to each model. The meaning of the four colors is as follows: Species belonging to the healthy state of the organism are colored blue, those belonging to the immune response are colored green, those belonging to infection like infected cells and viruses are colored magenta, and bacteria from bacterial co-infection are colored orange. Horizontally, the diagram consists of four lines. The models in the lowest line contain organizations with exactly two different kinds of species (colors) (including the empty set). In the second line above, there are three different combinations of species (colors) to be found in each model. There is only one model in each of the highest two lines: The Smith model [4] is the only one with bacteria and contains four different combinations of colors. In the Handel Model, there are even five different combinations of colors out of $2^{4}=16$ possible combinations.

**Table 1 viruses-11-00449-t001:** Overview of all models and organization types contained. An organization type like XX denotes the type of species contained in an organization, according to our coloring scheme. The set of organization types of a model is called its signature.

Model	Number of Variables	Number of Reactions	Number of Organizations	Organizations & Signature			
Baccam [13] 2006	3	4	*2*	$O_{1}^{Ba1}=\emptyset$			
				$O_{2}^{Ba1}$	**X**		
Miao [14] 2010	3	5	*3*	$O_{1}^{M}=\emptyset$			
				$O_{2}^{M}$	**X**		
				$O_{3}^{M}$	**X**	**X**	
Baccam II [13] 2006	4	5	*2*	$O_{1}^{Ba2}=\emptyset$			
				$O_{2}^{Ba2}$	**X**		
Pawelek [23] 2012	5	9	*2*	$O_{1}^{P}=\emptyset$			
				$O_{2}^{P}$	**X**		
Smith [15] 2016	5	12	*4*	$O_{1}^{Sm}=\emptyset$			
				$O_{2}^{Sm}$	**X**		
				$O_{3}^{Sm}$	**X**		
				$O_{4}^{Sm}$	**X**	**X**	
Handel [24] 2010	7	12	*5*	$O_{1}^{Ha}=\emptyset$			
				$O_{2}^{Ha}$	**X**		
				$O_{3}^{Ha}$	**X**		
				$O_{4}^{Ha}$	**X**	**X**	
				$O_{5}^{Ha}=\{all\}$	**X**	**X**	**X**
Hernandez [26] 2012	7	16	*2*	$O_{1}^{He}$	**X**	**X**	
				$O_{2}^{He}=\{all\}$	**X**	**X**	**X**
Cao [27] 2015	7	26	*3*	$O_{1}^{C}=\emptyset$			
				$O_{2}^{C}$	**X**		
				$O_{3}^{C}=\{all\}$	**X**	**X**	**X**
Saenz [28] 2010	8	12	*4*	$O_{1}^{Sa}=\emptyset$			
				$O_{2}^{Sa}$, $O_{3}^{Sa}$, $O_{4}^{Sa}$	**X**		
Hancioglu [29] 2007	10	44	*3*	$O_{1}^{Hcg}$	**X**		
				$O_{2}^{Hcg}$	**X**	**X**	
				$O_{3}^{Hcg}=\{all\}$	**X**	**X**	**X**
Bocharov [30] 1994	10	45	*2*	$O_{1}^{Bo}$	**X**		
				$O_{2}^{Bo}$	**X**	**X**	
Lee [9] 2009	15	37	*8*	$O_{1}^{L}$	**X**		
				$O_{2}^{L}$, $O_{3}^{L}$, $O_{4}^{L}$, $O_{5}^{L}$, $O_{6}^{L}$	**X**	**X**	
				$O_{7}^{L}$, $O_{8}^{L}=\{all\}$	**X**	**X**	**X**

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
