# Peer review of "Structure and Hierarchy of Influenza Virus Models Revealed by Reaction Network Analysis"

_viruses, 2019, doi:10.3390/v11050449_

Round 1
Reviewer 1 Report
I think this manuscript present a very nice work of theorization. The model is very well built in materials and methods. The theory is not too complicated to understand for a bioinformaticien but more difficult for a biologist. There may be a lack of understanding. This is why, although your general biological introduction is complete and explicit, I think that a biological explanation of the theoretical model is missing at the end of this introduction.
The manuscript is more like a review than an article since the only result is the hierarchy of the models created by other teams. If the authors propose new models since they suggest that they should exist in paragraph 3.12, this manuscript will eventually have the form of an article.
In the results and discussion section, the explanation that allows you to switch from one model to another is very clear and easily understood once the M & M read. In the paragraph 3.12: The formalism of the organizations is well explained. This chapter really helps to order the different models relative to each other. In the conclusion chapter the authors explain that this work will make it easier to choose the model corresponding to a particular situation and I agree with them.
Major:
The authors claims that the study presented in this manuscript shows that new models could be implemented and I think it would be interesting to do it. Otherwise this manuscript would look more like a review than a research article.
Minor:
1. At the end of the introduction, you should add a paragraph that makes a more obvious link between the reality of an infection for a virologist and your mathematical model. To clearly define that you are modelising an infection and for that you must first define basic concepts that are:
o Infected and uninfected cells as well as viruses are grouped under the name species.
o Reactions are examples of cell infection by a viruses or generation of new viruses from an infected cell or ....
o ……
Say that network pictures are diagrams of these reactions and give an example of organization. Say why you are going to need a matrix. Say why you need to introduce parameters β, δ, p and c.
All of this is necessary in order for biologists to understand the mathematical models that follow.
2. The term "chemical organization" may be better replaced by "biological organization".
3. Line 133-134: can you explain what are "periodic and chaotic attractors".
4. Line 179: say that Fig. A4 is in the appendix.
5. In the paragraph 3.12: do organizations' signatures characterize all models?
6. Line 412: the authors say that their approach could be applied to other viruses, but in my opinion they should necessarily be enriched, especially to follow the treatments that will be significantly different but also that will imply the implementation of other species of other reactions and the addition of new molecules.
7. Page 19: there is an appendix A but no an appendix B.
Reviewer 2 Report
The manuscript analysed and classified a number of published dynamic models of within-host influenza infection using a method of reaction network analysis. The motivation is good because with an increasing volume of models in the literature model classification and selection become more and more important. However, I do not think the method used in the manuscript is appropriate to study those influenza models for three reasons. The first is that the method cannot provide useful insight into the transient dynamics of the models which are shown to be more important for short-term influenza infection (as opposed to HIV which is highly determined by steady state). The second, I have no idea how the proposed hierarchy of the models (Figure 13) can be useful to facilitate model selection. In my opinion, the most important thing is how the underlying processes are modelled (e.g. with or without immunity, with or without delay, linear or non-linear, etc.) and what predictions they would make (in terms of some viral characteristics such as peak viral load, clearance rate, infection time, etc.). Finally, I cannot see any benefit of using the chemical organisation theory compared to the established stability and bifurcation theory for analysing dynamical systems. The chemical organisation theory may be arguably a better way of analysing a system with a large number of variables and parameters, but I am not convinced that I should use it to study those influenza models.
I suggest major revision for this manuscript because I may be biased and would like to hear more from the authors about the importance of the work. In particular, I hope the authors can clearly explain (1) what we can learn from the study in addition to what we have known from stability analysis and (2) how the new knowledge can potentially advance our understanding of biological problems.
Minor issues:
Line 36: Please clarify “completely and in a straightforward way”. Do you mean using fundamental theory of stability analysis or something else?
Line 57: We normally call it the target cell limited model.
Round 2
Reviewer 2 Report
The manuscript has been significantly improved. Particularly the authors demonstrate the usefulness of the novel analysis. I am happy to accept the manuscript after some minor comments are addressed:
- Duplicate reference: 23 and 29.
- Two following papers should be cited because they raised the importance of model selection for modelling and predicting influenza A infection, which is essentially a key motivation for developing a method to better understand and classify existing dynamic models (e.g. as mentioned in the introduction line 54-55). They may also be briefly discussed in the Conclusion.
1. Handel et al. 2007. Neuraminidase inhibitor resistance in influenza: Assessing the danger of its generation and spread. PLoS Comput Biol
2. Cao et al. 2017. The mechanisms for within-host influenza virus control affect model-based assessment and prediction of antiviral treatment. Viruses.
Author Response
We thank the reviewer for his/her comments.
We have cited both articles:
1- adding new sentence "... complementing established quantitative selection methods, such as those using the area under the viral load curve \cite{cao2017}"
2- cited Handel 2007 with model's section on Handel 2009 model.
